# Estimating the Time Toxicity of Contemporary Systemic Treatment Regimens for Advanced Esophageal and Gastric Cancers

**DOI:** 10.3390/cancers15235677

**Published:** 2023-11-30

**Authors:** Neha Y. Agrawal, Rajat Thawani, Corbin P. Edmondson, Emerson Y. Chen

**Affiliations:** 1Department of Medicine, Oregon Health & Science University, Portland, OR 97239, USA; agrawaln@ohsu.edu (N.Y.A.);; 2Department of Medicine, Section of Hematology/Oncology, University of Chicago, Chicago, IL 60637, USA; 3Division of Hematology and Medical Oncology, Knight Cancer Institute, Oregon Health & Science University, Portland, OR 97239, USA

**Keywords:** esophageal cancer, gastric cancer, time toxicities, hospital days, clinical trials

## Abstract

**Simple Summary:**

The treatment for advanced and metastatic esophagogastric malignancies has historically required time-intensive multi-agent chemotherapy regimens, where patients spend time away from home to engage in therapy administrations and supportive care. With the addition of immunotherapies to the standard of care, the authors completed this study to estimate the time spent in health care across various systemic treatment regimens. The authors estimated the time toxicity, or the days spent in health care facilities, due to cancer diagnosis and treatment with immunotherapy- and chemotherapy-based treatment regimens for esophagogastric cancers.

**Abstract:**

(1) Background: The purpose of this study was to evaluate the time toxicity, or time spent in health care, of immunotherapy- versus chemotherapy-based regimens for metastatic esophageal and gastric cancers. (2) Methods: A literature search was conducted, and 18 phase III clinical trials of immune checkpoint inhibitors were selected for analysis. Health care days were calculated based on the number of days associated with receiving therapy and the adverse events reported in the clinical trials. Both the number of health care days and the median overall survival were compared among chemotherapy-only, immunotherapy-only, and chemo-immunotherapy regimens across this cohort of drug registration trials. (3) Results: The estimated median number of health care days was 37 (range of 7–52) days, or 1.2 (range of 0.2–1.7) months, compared to a median survival of 10.2 months across these 18 studies. For the chemotherapy-only regimens, the median number of health care days was 39 (range of 21–51) days, and for chemo-immunotherapy, it was 39 (range of 30–52) days. The immunotherapy-only regimens had fewer days, a median of 28 (range of 24–41), *p* < 0.05, compared to the other two arms. (4) Conclusions: The chemo-immunotherapy regimens did not add time toxicity compared to chemotherapy alone. The immunotherapy-only regimens had lower time toxicity compared to chemotherapy alone. In the setting of decreased time toxicity and improved overall survival, further development of immunotherapy-based regimens could improve outcomes in advanced esophageal and gastric cancers.

## 1. Introduction

Unresectable and metastatic esophagogastric cancers frequently result in 5-year survival outcomes of less than 20 percent [1,2]. Immune checkpoint inhibitors in combination with chemotherapy have become the preferred contemporary therapies for first-line metastatic esophagogastric cancers [3]. In the setting of poor prognosis, palliative systemic options could control symptoms, slow progression, and prolong survival, though at the cost of side effects, financial resources, and time commitment.

The concept of ‘time toxicity’—the amount of time associated with obtaining cancer care, in metastatic esophageal and gastric cancer—is becoming relevant in health outcomes research. It is typical to discuss the benefits, costs, and side effects of treatment; however, the time commitment for patients to engage in their care is seldom explored [4]. In particular, patients with metastatic esophagogastric cancers are confronted with medical decisions in the context of finite survival, making those decisions all the more important [5]. Despite time spent being a common concern for patients, clinicians rarely have the specific information to counsel patients on the time burden of care. Quantifying the time associated with treatment methodically without direct patient-level data is challenging [6]. Gupta et al. proposed using “days with physical health care system contact” as a comprehensive measure of time toxicity, which includes clinic visits, infusions, procedures, bloodwork, urgent care visits, and overnight stays [7]. Other more specific measures have included hours recorded by digital technologies, patient surveys, electronic medical record time stamps, and clinical trial case logs. Studies on breast cancer using process mapping, administrative claims, and patient surveys have already demonstrated that cancer-related healthcare leads to significant inpatient time and time burdens for patients [8]. In a retrospective study of 23,382 Medicare beneficiaries, it was found that a longer length of time spent traveling to receive cancer care led to a higher risk of inpatient hospitalizations and more time engaging with the health care system [9]. Furthermore, specifically in gastric cancer, estimates of the time associated with medical care based on Medicare claims data showed that in the last year of life, 512 h (about 3 weeks) were committed to health care activities [10]. Distributing these hours over their corresponding days in the last year of life, the total time spent on medical care, or time toxicity by number of days, is likely much longer.

This concept of time toxicity may help patients make informed decisions about their treatment options, as certain regimens are more time-intensive than others despite comparable survival benefits. For instance, patients would benefit from knowing not just how much survival benefit may be gained from treatment, but also the volume of time lost in the hospital or at outpatient facilities. This dynamic was explored by Hall et al., who found that patients tended to distinguish between existential time (quality time left to live) and chronologic time (time spent engaging in cancer-related care), and as a result, they found the burden of time spent on cancer had a negative impact on patient well-being [11]. However, there is a paucity of research that accurately describes the amount of time spent in cancer care for esophageal and gastric cancers. Additionally, contemporary drug registration trials do not publicly share or accurately document patient-level data regarding the hours and days participants spend in the study.

To our knowledge, this is the first pilot study that estimates time toxicity for chemotherapy and immunotherapy regimens in esophagogastric malignancies. While chemo-immunotherapy is currently the standard of care for most metastatic esophagogastric cancer-based drug registration trials, dual checkpoint inhibitors and newer immunotherapy-only (chemotherapy-free) regimens are increasingly being tested [12]. Time toxicities associated with cytotoxic chemotherapy, immunotherapy, and chemo-immunotherapy could be factored into decision-making regarding the initiation of standard-of-care therapy. The aim of our study was to calculate the health care days associated with treatment as a measurement of time toxicity in the standard treatment of unresectable and metastatic esophagogastric malignancies to determine if the time toxicity varies among chemotherapy alone, immunotherapy alone, and chemo-immunotherapy combinations.

## 2. Methods

### 2.1. Overview

A literature review to identify phase III clinical trials that evaluated immunotherapy-containing regimens for patients with metastatic esophageal and/or gastric cancer was conducted. These contemporary studies were included to examine the overall survival of patients receiving treatment and to explore the potential contribution of time toxicity to the treatment of esophagogastric malignancies.

### 2.2. Literature Search and Selection

The authors performed a detailed search using https://clinicaltrials.gov/, (accessed on: 6 June 2023) a public online database of registered interventional trials that is managed by the National Library of Medicine and National Institutes of Health. The study selection was based on a prior literature search and analysis [13]. The search words included were “esophageal cancer”, “gastric cancer”, and FDA-approved anti-PD1 and anti-PDL1 immunotherapies. Tislelizumab was included, as it remains under FDA review for potential future approval. The results were subsequently filtered so that only randomized controlled trials were selected. Trials with published analysis by 6 June 2023 were included for data extraction. Two reviewers conducted the literature search and agreed upon the selected trials (E.Y.C., N.Y.A.). Similar methods used in the prior analysis were followed with updated data cutoffs and endpoint-related time toxicities.

### 2.3. Data Extraction

For each published trial, the authors collected data about the study drug of interest, study population, study design, schedule of events, adverse events, clinical efficacy, and survival endpoints. The published methods and protocols of each clinical trial were reviewed to estimate the number of ‘health care days.’ Given the difficulty in quantifying the exact time spent with lab draws, travel to the hospital, clinic visits, and hospitalizations associated with cancer care, each encounter with the healthcare system was counted as a health care day, or ‘hospital day’ in the related literature. Health care days were identified based on the trial data by identifying the number of screening days at the initiation of the trial, infusion days for treatment-related care, and imaging surveillance days. The reported proportion of serious and severe adverse events from treatment was also included as a surrogate for outpatient and inpatient time associated with adverse events. To do this, a proportion of grade 3 or higher serious adverse events for each treatment arm was multiplied by 7.6 days, the average length of hospital stays reported for patients with esophageal cancer according to the published literature [14]. One health care day was assigned for non-serious grade 3 or higher adverse events and serious grade 1 or 2 adverse events, as they likely would be addressed by outpatient or urgent care settings. The total health care days for every treatment regimen in all relevant clinical trials were calculated to form an estimate of time toxicity. All the extracted data were available online without patient-level data and did not contain any personal health information, so local institutional review board (IRB) submission was not applicable.

### 2.4. Statistical Analysis

Health care days were calculated for all randomized groups in the selected clinical trials. Comparisons of the time toxicities and survival outcomes were made among (1) chemotherapy-only, (2) immunotherapy-only, (3) chemotherapy with immunotherapy (chemo-immunotherapy), and (4) best supportive care arms, using either the Mann–Whitney or Kruskal–Wallis test. Descriptive calculations and statistical testing were completed using SAS version 9.4 (SAS Institute Inc., Cary, NC, USA). All figures were produced using Microsoft Excel and Microsoft PowerPoint (Microsoft Office Professional Plus 2016).

## 3. Results

Eighteen clinical trials were included in this study for time toxicity estimation (Figure 1). There were 38 randomized groups, as 2 trials had 3 randomized groups. Only pembrolizumab, nivolumab, avelumab, and tislelizumab are represented here (Table 1). Nine trials examined immune checkpoint inhibitors in the first-line metastatic setting; six were in the second line, two were in the third line, and one was an adjuvant trial. A total of 12,378 patients were represented in the 38 randomized groups. Sixteen arms were cytotoxic chemotherapy only, twelve were immunotherapy only, eight were chemo-immunotherapy, and two were placebo or best supportive care. The median overall survival among these studies combined was 10.2 months.

The estimated median number of health care days was 37 (range of 7–52) days, or 1.2 (range of 0.2–1.7) months per year. The highest number of health care days occurred with chemotherapy only (median of 39 days (range of 21–51)) and chemo-immunotherapy only (median of 39 days (range 30–52)), as noted in Table 2. The immunotherapy-only arms had a median of 28 days (range of 24–41). The longest median survival was observed in the chemo-immunotherapy arms, with a median of 14.4 months (range of 12.5–17.5) (Table 2). The median overall survival was shorter in the chemotherapy-only (8.8 months (range of 5.0–7.2)) and immunotherapy-only arms (9.1months (range of 4.6–13.7)), and the best supportive care arms (4.1 months, *p* < 0.01, Table 2). Among the metastatic trials, the median progression-free survival rates were 4.4 (2.1–8.4) months in chemotherapy only, 2.0 (1.4–4.0) months in immunotherapy only, and 7.3 (6.9–10.9) months in the chemo-immunotherapy arms. The median response rates were 21% (4–52%) in chemotherapy only, 16% (2–35%) in immunotherapy only, and 55% (45–74%) in the chemo-immunotherapy arms. The proportion of time dedicated to health care compared to the median overall survival was highest in the chemotherapy-only arms (15%) and lowest in the supportive care arms (7%, Table 2).

The immunotherapy-only arms had shorter health care days, or time toxicity, when compared to the chemotherapy-only arms (*p* = 0.02) and chemo-immunotherapy arms (*p* < 0.05, Figure 2). However, there was no difference in the health care days between the chemotherapy and chemo-immunotherapy arms (Figure 2). The details of all 18 trials with 38 randomized arms, with their referenced PubMed PMID, are listed in Table 3.

## 4. Discussion

Even under an optimal workflow among patients with good functional status who have metastatic esophageal and gastric cancer, approximately 37 days, or 1.2 months, are dedicated to medical care, when their overall survival could be less than 12 months. The data here suggest that treatments that provide limited overall survival may be less meaningful in the context that patients will spend a significant amount of time in the hospital or outpatient treatment centers. One study of 18,486 patients noted that 92% of patients receiving chemotherapy for metastatic cancer were hospitalized at least once, and 51% were hospitalized for a complication likely directly related to chemotherapy [15]. Other studies have also attempted to elucidate where patients with cancer spend their time, and similarly have concluded that a sizable portion of time is spent receiving oncologic care, which can be challenging in the setting of life-limiting disease [9,16,17,18]. One such study discovered the average amount of time a patient spent on a single oncology appointment was approximately 3 h, and only 20 minutes were allotted for direct patient-provider interaction [19]. Despite these challenges, the use of telehealth has increased during the COVID pandemic, which could reduce the time toxicity burden from cancer-related care and improve patient well-being [20,21,22].

Patients who received chemotherapy versus chemotherapy-free (or immunotherapy only) treatment had a significantly greater time commitment to medical care, by a difference of approximately nine health care days. Such a difference remains especially meaningful if the overall survival difference between the treatment arms in clinical trials is only several weeks to fewer than 3 months. In the real world, where patients suffer from more co-morbidities and cancer-related symptoms, the number of health care days spent is likely even greater, thereby possibly diminishing, or even negating, the survival benefit [23]. Chemo-immunotherapy has demonstrated better overall survival than chemotherapy alone, yet the time toxicities remained the same. While chemo-immunotherapy has the highest level of evidence for clinical efficacy, it may also have more risks and time toxicities. Drug development should focus on developing more effective immunotherapy combinations, such as dual checkpoint inhibition, rather than relying on conventional cytotoxic chemotherapy drugs that require intensive supportive care, long infusion times, and frequent monitoring. Overall, the results of this study suggest that immunotherapy-based treatment decreases time toxicity and enhances quality of life by preserving non-medical time.

Most of the time toxicity in this analysis was from days spent on lab evaluations, imaging, provider assessments, and infusion appointments. Serious adverse events were a minority of the time spent in the hospital due to the infrequency at which they occurred in the clinical trial patients. In the real world, patients may develop more side effects, have serious co-morbidities, and experience more complications that require in-depth workups, consultant referrals, and hospitalizations. This study can only make logical estimates based on the available adverse event reporting and the related literature, so estimates may be lower than expected. Reviewing electronic medical charts directly may be a more accurate method to estimate the time toxicity days versus home days by counting the number of visits from outpatient clinics, laboratories, imaging, emergency departments, and hospitalizations [24]. Some chemo-immunotherapy regimens, such as nivolumab and FOLFOX every two weeks, inherently use many health care days. According to the schedule of events in the protocol, 52 health care days are associated with therapy alone over the course of one year [25]. Future systemic treatment developments could decrease the use of continuously infused drugs and increase the duration of treatment cycles to prevent unnecessary time spent in health care facilities. The isolated perfusion of chemotherapy to the liver, lung, or other organs could be researched. Decreasing the use of cisplatin may also decrease the need for hydration appointments, electrolyte repletion, and complications that could lead to hospitalizations. In contrast, study arms with immune checkpoint inhibitors, whether single or double, generally have fewer days needed for drug administration, particularly if the infusion is only every three or four weeks.

There are several study limitations here. The number of phase III trials for chemo-immunotherapy is relatively small in esophagogastric cancer, and this analysis cannot be extrapolated to other cancer types. The authors here encourage other researchers to repeat similar analyses in their respective disease focus. The surrogate of health care days for time toxicity is based on a conservative estimate based on published study protocols and reported adverse events, as these clinical trials did not report the duration of hospitalizations or the details of specific adverse events. The authors encourage clinical trial sponsors to make patient-level data available for researchers to conduct more in-depth analyses. Further transparency surrounding the reporting of serious adverse events and their hospitalization lengths can lead to more robust estimates of the time toxicity. Finally, this study measured the time toxicity based on the patient’s perspective and neglected the time spent by caregivers who take time away from their personal lives and miss work and family events to engage in care for their loved ones. One study noted that caregivers provide, on average, 8 h of informal caregiving per day to cancer survivors [26]. These limitations point to the need for more research on time toxicity to better understand the degree to which time toxicity could inform treatment decisions for patients and their families.

## 5. Conclusions

In summary, chemotherapy-free regimens exhibit less time toxicity compared to chemotherapy-only regimens. Chemo-immunotherapy has demonstrated superior clinical efficacy compared to chemotherapy alone without increasing time toxicities. The results of this study show the importance of developing novel immunotherapy combinations that could be less time consuming. Time is indeed valuable to patients with advanced cancer, and time spent in the hospital should be factored into treatment decisions in the setting of life-limiting disease to improve their quality of life. While the balance between clinical efficacy and time toxicity may vary from one perspective to another, an accurate estimate can nevertheless provide patients with valuable data points to consider when receiving cancer care. Examining the study protocols, treatment schedules, and extent of adverse events could be used to estimate health care days as a surrogate for time toxicity. The application of time toxicity evaluation can be factored into value-based frameworks, just as we evaluate the clinical benefit, financial toxicity, and treatment toxicities when discussing personalized cancer care.

## Figures and Tables

**Figure 1 cancers-15-05677-f001:**
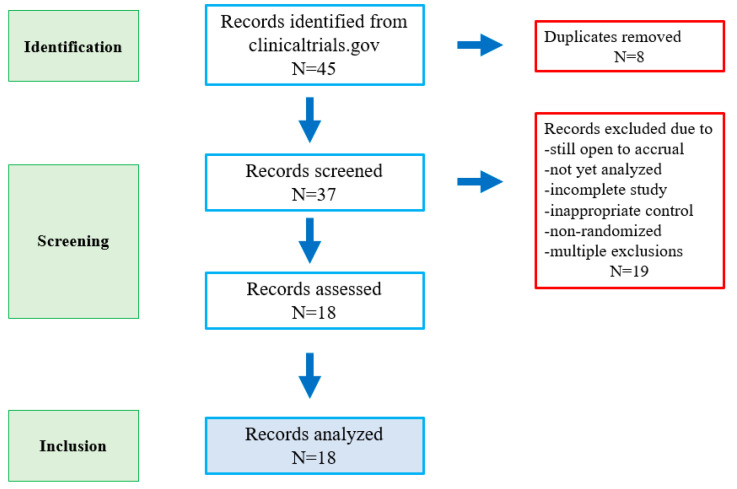
Flow diagram of studies analyzed.

**Figure 2 cancers-15-05677-f002:**
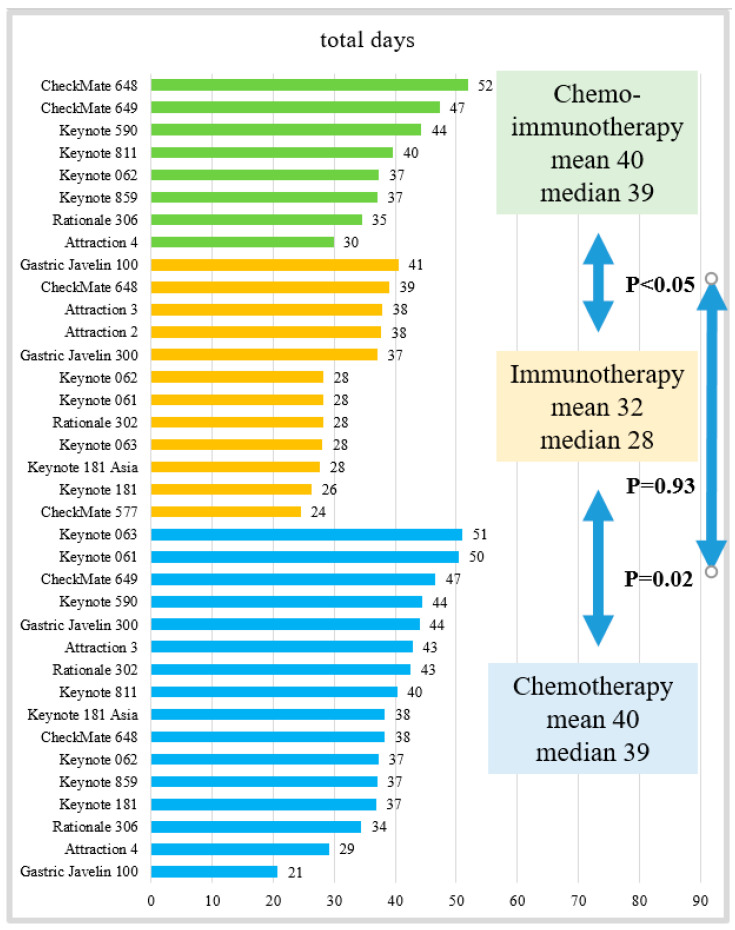
Time toxicities of chemo-immunotherapy, immunotherapy, and chemotherapy arms (N = 36, excluding supportive care arms).

**Table 1 cancers-15-05677-t001:** Clinical trial characteristics (N = 18).

Characteristics	N (%)
Immunotherapy Type	
Pembrolizumab	8 (45%)
Nivolumab	6 (33%)
Avelumab	2 (11%)
Tislelizumab	2 (11%)
Cancer Type	
Gastric and GE junction	9 (50%)
Esophageal	8 (44%)
Esophageal, GE junction, and gastric	1 (6%)
Line of Therapy	
First-line metastatic	9 (50%)
Second-line metastatic	6 (33%)
Third-line metastatic	2 (11%)
Adjuvant	1 (6%)
Sample size (median, range)	639 (94–1581)
Treatment Arms (N = 38)	
Chemotherapy only	16 (42%)
Immunotherapy only	12 (32%)
Chemotherapy with immunotherapy	8 (21%)
Best supportive care	2 (5%)
Median overall survival (median, range; N = 34)	10.2 (4.1–17.5) months
Median progression-free survival (median, range; N = 34)	4.0 (1.4–10.9) months
Objective response rate (median, range; N = 36)	21% (0–74%)

**Table 2 cancers-15-05677-t002:** Time toxicity and median overall survival of selected therapies.

	Time Toxicity (Days)	Median Overall Survival (Months)	Proportion of Survival
	Median (Range)	Median (Range)	%
Chemotherapy only (N = 16)	39 (21–51)	8.8 (5.0–17.2) ^a^	15%
Immunotherapy only (N = 12)	28 (24–41)	9.1 (4.6–13.7) ^b^	9%
Chemotherapy with immunotherapy (N = 8)	39 (30–52)	14.4 (12.5–17.5) ^c^	9%
Best supportive care (N = 2)	9 (7–11)	4.1 (4.1–4.1) ^d^	7%
*p* value	<0.01	<0.01	

^a^ N = 15; ^b^ N = 11; ^c^ N = 7; ^d^ N = 1.

**Table 3 cancers-15-05677-t003:** Key details of anti-PD-1 and anti-PD-L1 immunotherapies in phase III trials.

Study Name	Registration # (Pub Med PMID)	Cancer Type	Experimental Arm	Control Arm
Pembrolizumab				
KEYNOTE 590	NCT03189719 (34454674)	Esophageal (all histology), 1st-line advanced/metastatic	5-Flourouracil (5FU) and cisplatin with pembrolizumab	5FU and cisplatin with placebo
KEYNOTE 181 Asia	NCT03933449 (34973513)	Esophageal (all histology), 2nd-line advanced/metastatic	Pembrolizumab	Docetaxel, paclitaxel, or irinotecan
KEYNOTE 181	NCT02564263 (33026938)	Esophageal (all histology), 2nd-line advanced/metastatic	Pembrolizumab	Docetaxel, paclitaxel, or irinotecan
KEYNOTE 061	NCT02370498 (29880231)	Gastric/GE junction adenocarcinoma, 2nd-line advanced/metastatic	Pembrolizumab	Paclitaxel
KEYNOTE 063	NCT03019588 (34878659)	Gastric/GE junction adenocarcinoma (PD-L1 CPS ≥ 1), 2nd-line advanced/metastatic	Pembrolizumab	Paclitaxel
KEYNOTE 811	NCT03615326 (34912120)	Gastric/GE junction adenocarcinoma (HER2+), 1st-line unresectable/metastatic	Chemotherapy with trastuzumab and pembrolizumab	5FU and cisplatin, or capecitabine and oxaliplatin (CAPOX), with trastuzumab
KEYNOTE 062	NCT02494583 (32880601)	Gastric/GE junction adenocarcinoma (HER2−, PD-L1 CPS ≥ 1), 1st-line advanced/metastatic	Chemotherapy with pembrolizumab (also has pembrolizumab monotherapy arm)	5FU or capecitabine, and cisplatin, with placebo
KEYNOTE 859	NCT03675737 (37293712)	Gastric/GE junction adenocarcinoma (HER2−)	FP or CAPOX with pembrolizumab	FP or CAPOX with placebo
Nivolumab				
CHECKMATE 648	NCT03143153 (35108470)	Esophageal, (squamous), 1st-line unresectable, recurrent, or metastatic	5FU and cisplatin with nivolumab (also has nivolumab with ipilimumab arm)	5FU and cisplatin
CHECKMATE 649	NCT02872116 (34102137)	Esophageal, GE junction, gastric (HER2− adenocarcinoma), 1st-line unresectable/metastatic	5FU or capecitabine, and oxaliplatin (FOLFOX or CAPOX), with nivolumab	FOLFOX or CAPOX
ATTRACTION 4	NCT02746796 (35030335)	GE junction, gastric (HER2− adenocarcinoma), 1st-line recurrent/advanced	Capecitabine or S-1 and oxaliplatin (CAPOX or SOX) with nivolumab	CAPOX or SOX
ATTRACTION 3	NCT02569242 (31582355)	Esophageal (squamous), 2nd-line advanced/metastatic	Nivolumab	Docetaxel or paclitaxel
ATTRACTION 2	NCT02267343 (28993052)	Gastric/GE junction adenocarcinoma, 3rd-line recurrent/metastatic	Nivolumab	Placebo
CHECKMATE 577	NCT02743494 (33789008)	Esophageal (all histology), adjuvant after tri-modality therapy for resectable cancer	Nivolumab	Placebo
Avelumab				
Javelin Gastric 100	NCT02625610 (33197226)	Gastric/GE junction adenocarcinoma, maintenance after induction 1st-line chemotherapy	Avelumab	Continuation of FOLFOX or CAPOX
Javelin Gastric 300	NCT02625623 (30052729)	Gastric/GE junction adenocarcinoma, 2nd-line unresectable, recurrent or metastatic	Avelumab	Irinotecan or paclitaxel
Tislelizumab				
Rationale 302	NCT03430843 (35442766)	Esophageal squamous cell carcinoma, 2nd-line unresectable or metastatic	Tislelizumab	Paclitaxel, docetaxel, or irinotecan
Rationale 306	NCT03783442 (37080222)	Esophageal squamous cell carcinoma, 1st-line unresectable or metastatic	Cisplatin or oxaliplatin plus fluorouracil (5-FU); cisplatin or oxaliplatin plus capecitabine; cisplatin or oxaliplatin plus paclitaxel, with tislelizumab	Cisplatin or oxaliplatin plus fluorouracil (5-FU); cisplatin or oxaliplatin plus capecitabine; cisplatin or oxaliplatin plus paclitaxel, with placebo

## Data Availability

The data presented in this study are openly available in Table 3, where all references with their PubMed ID are listed.

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
