# Peer review of "Estimating the Time Toxicity of Contemporary Systemic Treatment Regimens for Advanced Esophageal and Gastric Cancers"

_cancers, 2023, doi:10.3390/cancers15235677_

Round 1

Reviewer 1 Report

Comments and Suggestions for Authors

The authors estimated the time toxicity, or the days spent in health care facilities, due to cancer diagnosis and treatment with immunotherapy and chemotherapy-based treatment regimens for esophagogastric cancers. The concept of time toxicity is interesting, and this manuscript is well constructed with some information.

Author Response

Thank you very much for your kind words. We agree this is a novel concept that has recently been brought to light in the oncology literature, and so we strive to focus on specific cancer types and therapies as the next step. We appreciate your taking the time to review and consider the manuscript for publication.

Reviewer 2 Report

Comments and Suggestions for Authors

This study is interesting and has an original idea. Indeed, it is clear from recent studies that immunotherapy leads to fewer side effects than chemotherapy. However, it is important to emphasize that clinical efficacy and overall survival are equally critical factors in treatment selection. The analysis could have been more robust by including a more in-depth evaluation of clinical efficacy.

I believe it is risky to compare immunotherapy alone versus immuno-chemo/chemo in such heterogeneous patient populations and pathologies. The conclusions of the paper in its current form appear somewhat speculative. Perhaps, removing the immunotherapy-only group might be a consideration, but this would require major revisions.

Reviewer 3 Report

Comments and Suggestions for Authors

In these considerations of time toxicity,do the authors see a role for regional chemotherapies such as intra-arterial infusion or isolated perfusion in later analyses of other tumor entities due to the usually significantly lower toxicity and sometimes very good efficacy?

Without going into further detail,Ithink this work is very good and worth publishing because it deals critically with the mostly neglected and overlooked problem of cancer treatment with regard to quality of life in general.

Author Response

Fascinating perspective! Intra-arterial infusion or isolated perfusion are still considered experimental for metastatic esophageal-gastric cancer, and such approach may still cause side effects when using cytotoxic chemotherapy like floxuridine. Some of the approaches may also be very time-intensive and may require additional surgeries and procedures with additional risks, such as hepatic arterial infusion or HIPEC approaches. Regardless, we agree with you that drug delivery/approach is as important as specific drug development in future research, and so we added one sentence in Discussion Paragraph 3 accordingly.

We appreciate your comments and time taken for reviewing the manuscript. We aim to look at other facets of cancer therapies not popularly explored.

Round 2

Reviewer 2 Report

Comments and Suggestions for Authors

The authors have taken into account the main comments made on the paper. The limitations are better explained. I propose to publish this paper in its current form.